# Bugs That Can Resist Antibiotics but Not Men: Gender-Specific Differences in Notified Infections and Colonisations in Germany, 2010–2019

**DOI:** 10.3390/microorganisms9050894

**Published:** 2021-04-22

**Authors:** Michael Brandl, Alexandra Hoffmann, Niklas Willrich, Annicka Reuss, Felix Reichert, Jan Walter, Tim Eckmanns, Sebastian Haller

**Affiliations:** 1Postgraduate Training for Applied Epidemiology, Department of Infectious Disease Epidemiology, Robert Koch Institute, 10113 Berlin, Germany; ReichertF@rki.de (F.R.); WalterJ@rki.de (J.W.); 2European Programme for Intervention Epidemiology Training (EPIET), European Centre for Disease Prevention and Control (ECDC), 17183 Stockholm, Sweden; 3Unit for Healthcare-Associated Infections, Surveillance of Antibiotic Resistance and Consumption, Department of Infectious Disease Epidemiology, Robert Koch Institute, 10113 Berlin, Germany; HoffmannA@rki.de (A.H.); WillrichN@rki.de (N.W.); ReussA@rki.de (A.R.); EckmannsT@rki.de (T.E.); HallerS@rki.de (S.H.)

**Keywords:** antimicrobial resistance, gender distribution, methicillin-resistant *Staphylococcus aureus*, carbapenem-non-susceptible Enterobacterales, carbapenem-non-susceptible *Acinetobacter*

## Abstract

Data from surveillance networks show that men have a higher incidence rate of infections with anti-microbial-resistant (AMR) pathogens than women. We systematically analysed data of infections and colonisations with AMR pathogens under mandatory surveillance in Germany to quantify gender-specific differences. We calculated incidence-rates (IR) per 100,000 person–years for invasive infections with Methicillin-resistant *Staphylococcus aureus* (MRSA), and for infections or colonisations with carbapenem-non-susceptible *Acinetobacter* spp. (CRA), and Enterobacterales (CRE), using the entire German population as a denominator. We limited the study periods to years with complete notification data (MRSA: 2010–2019, CRA/CRE: 2017–2019). We used Poisson regression to adjust for gender, age group, federal state, and year of notification. In the study periods, IR for all notifications were 4.2 for MRSA, 0.90 for CRA, and 4.8 for CRE per 100,000 person-–years. The adjusted IR ratio for infections of men compared to women was 2.3 (95% confidence interval [CI]: 2.2–2.3) for MRSA, 2.2 (95%CI: 1.9–2.7) for CRA, and 1.7 (95%CI: 1.6–1.8) for CRE. Men in Germany show about double the risk for infection with AMR pathogens than women. This was also true for colonisations, where data were available. Screening procedures and associated hygiene measures may profit from a gender-stratified approach.

## 1. Introduction

Anti-microbial resistance (AMR) was identified as one of ten threats to human life by the World Health Organization (WHO) in 2019 [1]. An estimated 33,000 people die each year due to an infection with antibiotic-resistant bacteria in the European Union and European Economic Area [2]. Quantification of the true global burden of AMR would be essential to allocate adequate resources and inform public health action, but proved to be difficult due to a lack of high-quality and population-representative data [3,4]. In a working paper, the WHO called for applying the “gender lens” to analysis of surveillance data in order to successfully deal with AMR [5].

Data from the European Antimicrobial Resistance Surveillance Network (EARS-Net) show that men have a higher incidence of infections with the majority of the eight bacterial species under surveillance [6]. Analogously, studies have found a positive association between male gender and infections with AMR pathogens [7,8,9]. However, most of the studies detected gender differences as a by-product, and thorough analyses on this topic are scarce. Furthermore, regional effects between and within countries regarding both AMR prevalence and risk factors have been described and might be, in part, underlying causes of observed gender discrepancies [10,11,12].

We analysed all infections and colonisations with AMR pathogens that were notified in Germany since 2010 according to Section 7 of the Infection Protection Act [13]. We aimed to assess the magnitude of gender-specific differences in order to guide prevention and treatment measures. Exact estimates of this effect will contribute to the improvement in models and algorithms in predicting risks of infections with antibiotic-resistant bacteria. Eventually, men may represent a target group in the global fight against AMR.

## 2. Materials and Methods

We used data from the mandatory national surveillance system on the following pathogens:Invasive infections with Methicillin-resistant *Staphylococcus aureus* (MRSA), i.e., detection of MRSA in laboratory specimen from blood or cerebrospinal fluid;Infections or colonisations with *Acinetobacter* spp. (CRA) with carbapenem-non-susceptibility or detection of a carbapenemase determinant;Infections or colonisations with Enterobacterales (CRE) with carbapenem-non-susceptibility or detection of a carbapenemase determinant.

Study periods were limited to the full calendar years in which they were notifiable, i.e., for MRSA from 2010 to 2019, and for CRA and CRE from 2017 to 2019 [14]. We conducted a retrospective cohort study and used the entire German population as the study population, where every notification of one of the AMR pathogens was considered a case. With an approximate population of Germany at about 80 million inhabitants, this equals about 800 million person–years at risk for MRSA and 240 million person–years at risk for CRA and CRE.

We present distributions of notifications, as well as infections for CRA and CRE, by total numbers and proportions. From case numbers, we calculated incidence rates (IR) per 100,000 population and stratified by gender, age group, federal state, and year of notification. We used Poisson regression to calculate incidence rate ratios (IRR) for the risk of infection and colonisation with AMR pathogens of men compared to women. We, furthermore, present IR and IRR of age-stratified analysis. A *p*-value of lower than 0.05 was considered statistically significant.

For multivariable analysis, we used the stepwise forward model selection to adjust for confounding of the above-mentioned variables. Age was used as a categorical variable because of a non-linear relationship with the outcome. The goodness of fit was tested with a likelihood ratio test. We stratified the final multivariable model by infection vs. colonisation, hospitalisation status (inpatient vs. outpatient), and deceased vs. not deceased to account for effect modification.

Data on the German population were retrieved from The Federal Statistical Office (www.destatis.de/EN, accessed on 18 February 2021). This source also provided information on absolute numbers of inpatients by gender, age group, and federal state in 2018. From this, we calculated hospitalisation rates (HR) per 100,000 population and the HR ratio (HRR) of men compared to women. We used these data to control for hospital-acquired infections and additionally adjusted the data by excluding hospitalisations due to pregnancy, birth, and postpartum, as we considered the risk for women to be infected with an AMR pathogen in these particular settings to be neglectable. In the sensitivity analysis, we used the total numbers of inpatients (minus hospitalized women in relation to pregnancy) as a hypothetical population and reran the above regression models.

We further analysed possible regional differences by applying an ecological study design in which we examined the 401 districts in Germany in regard to pig density [15], population density [16], and the German Index of Socioeconomic Deprivation (GISD) [17]. We compared IRR for infections/colonisations of men to women in the 20 districts with the highest numbers of pig stables to the remaining German districts. Furthermore, we compared IRR in the 20 most densely populated (urban) to the 20 least densely populated (rural) districts and IRR in the 20 highest deprived to the 20 least deprived districts.

## 3. Results

In the study periods, there were 34,551 notifications of bloodstream infections with MRSA (median age 74 years, interquartile range (IQR) 64–80), as well as 2278 notifications with CRA (median age 66 years, IQR 53–76), and 12,055 with CRE (median age 68 years, IQR 55–78).

For CRA 568 (24.9%) infections, plus 1.061 (46.6%) colonisations were notified. For the remaining 649 (28.5%) notifications, these data were not collected or could not be determined. For CRE, 2265 (22.0%) infections and 5720 (47.5) colonisations were notified. On 3680 (30.5%) notifications, this information was undetermined or missing. Detailed characteristics of all notified cases can be found in Table 1.

We found an overall IR of 4.2 for MRSA, 0.90 for CRA, and 4.8 for CRE, per 100,000 person-years, respectively. In univariable analysis, the IRR of men compared to women for an invasive infection with MRSA was 1.8 (95% confidence interval [CI]: 1.7–1.8). For infection or colonisation with one of the other two examined pathogens, we observed an IRR of 1.9 (95%CI: 1.8–2.1) for CRA and 1.6 (95%CI: 1.6–1.7) for CRE (Appendix A).

For MRSA, we saw the highest IRR with 120 (95%CI: 100–150) in the over-80-year-olds compared to the reference group of 1–9-year-olds. For CRA, IR was maximal with 20 (95%CI: 14–29) in the 70–79 years age group. IRs for CRE were maximal in under 1-year-olds, with 15 (95%CI: 13–18), but of similar magnitude in the two oldest age groups. The differences between federal states were least strong for MRSA, with maximum IRRs in the federal states Mecklenburg-Vorpommern (IRR = 4.3, 95%CI: 3.7–4.9) and Saxony-Anhalt (IRR = 4.3, 95%CI: 3.8–4.9) compared to the reference state Bremen, where IR were among the lowest across all three pathogens. For both CRA and CRE, we observed maximum IRR in Berlin with 8.2 (95%CI: 4.8–14) for CRA and 4.9 (95%CI: 4.0–6.1) for CRE. The IR for bloodstream infections with MRSA increased until 2012 and subsequently decreased until 2019 to 0.47 (95%CI: 0.45–0.50), compared to 2010. From 2017 to 2019, the risk for infection or colonisation decreased to 0.90 (95%CI: 0.81–1.0) for CRA, but increased to 1.4 (95%CI: 1.3–1.4) for CRE. 

In age-stratified analyses, we observed the lowest IRR of men compared to women for infection or colonisation for all three pathogens in 1–9-year-olds (Figure 1). With increasing age, the IR for both men and women, but also the IRR of men to women, increased. For MRSA, the risk of infections for men, compared to women, was more than two-fold for all age groups over 50 years, with a maximum IRR of 2.4 (95%CI: 2.3–2.5) in age group 60–69. For CRA, we observed strongest fluctuations due to low case numbers and the highest IRR of 2.8 (95%CI: 1.8–4.2) in the 20–29 years age group. IR for CRE showed a similar course as for MRSA, and also peaked in age the 60–69 years age group, with an IRR of 2.2 (95%CI: 2.0–2.4).

In multivariable analyses of all notifications adjusted for age group, federal state, and year of notification, there was a roughly two-fold increase in the IR of men compared to women (Table 2). This was true for all three pathogens and exceeded results from univariable analysis. In stratified analysis, the IRR for infections with CRA was 2.2 (95%CI: 1.9–2.7), and with CRE it was 1.7 (95%CI: 1.6–1.8). For CRA, the effect size for colonisations was similar, while for CRE, the gender difference was significantly larger for colonisations (IRR = 2.0, 95%CI: 1.9–2.1). We observed a higher men-to-women IRR for inpatients compared to outpatients for both CRA and CRE, but not for MRSA. The IRR did not differ significantly for any of the three groups when stratifying by deceased vs. not deceased.

Looking at the numbers of full-time inpatients in Germany, in the year 2018, we saw an overall slight decrease in the HRR for men at 0.94 compared to women (Table 3). After the subtraction of hospitalisations related to pregnancy, birth, and postpartum, the association was the other way around and slightly elevated for men (HRR = 1.05). In age groups over 50, however, HR was clearly higher for men, with the highest rate difference of 32% in the group of 60- to 69-year-olds. 

In sensitivity analysis, when replacing the German population with the full-time inpatient population, excluding pregnancy-related hospitalisations, the IR of men still exceeded those of women. However, in age-stratified analysis of all notifications, the IRR of men to women in older age groups were considerably smaller than in the German population (Appendix A). In multivariable analysis, the adjusted IRR for infections of men compared to women were 1.8 (95%CI: 1.8–1.9) for MRSA, 1.9 (95%CI: 1.6–2.2) for CRA, and 1.4 (95%CI: 1.3–1.5) for CRE. The IRR for colonisations was significantly higher, with 1.7 (1.6–1.8) for CRE, but not for CRA. There were no statically significant differences when additionally restricting to inpatient cases in the surveillance dataset or between deceased and not-deceased for any of the pathogens.

In our ecological study, the districts with the highest densities in pigs did not show a significantly differing gender distribution in IR compared to notifications of remaining German districts (Appendix A). In the 20 most rural districts, we observed slightly higher IRR of men compared to women than in the 20 most urban districts for MRSA and CRE, but the effect was the opposite for CRA. Comparing the top districts to the bottom districts with regards to GISD revealed no effect on the gender distribution.

## 4. Discussion

We found that, for men, across age groups, federal states, and years under investigation, the risk for infections with investigated antibiotic-resistant bacteria was increased by factor 1.7 to 2.3. The effect of gender on colonisations was of a similar magnitude, with 2.3 and 2.0 for CRA and CRE, respectively. The IRR increased considerably with age, and highlights the potential for prevention of both colonisations and infections with AMR pathogens in older men. Previous studies that found a higher risk of infection with antibiotic-resistant pathogens for men came to varying conclusions: while some studies suggested underlying biological mechanisms as the cause [18], others proposed differences in antibiotic prescribing [19], or poorer compliance of men with hand-hygiene recommendations [9]. Furthermore, men were found to be more likely to be affected by hospital-acquired infections [20,21], which may be caused by higher hospitalisation rates, especially in older age groups. By using full-time inpatients as an alternate population, we deemed all infections and colonisations to be hospital-acquired. Although the calculated effect was reduced in magnitude, men still had an at least 40% increased risk for infection with any of the three pathogens. This underlines that the differences due to gender cannot only be attributed to differing hospitalisation rates. To explore whether length of exposition in the hospital contributed to the observed effect, we additionally compared length of hospital stay between men and women, but observed no noteworthy differences (data not shown). This reinforced our confidence in the stability of our results. 

To put the observed effect in AMR pathogens into perspective, we looked at the gender distribution of infections with susceptible bacteria in Germany in 2019 [14]. In particular, four other notifiable bacterial infections had similar or higher men-to-women ratios in incidence rates: syphilis (15), legionellosis (2.4), tuberculosis (2.0), and Q fever (1.9). For syphilis, the excess incidence rate in men can most likely be attributed to sexual transmission in men who have sex with men [22,23]. Respiratory infections with *Legionella* causing legionellosis were found to be associated with behavioural factors like smoking, and occupational factors like working with machinery, both exposing men to a higher risk of infection [24,25]. In patients with tuberculosis and Q fever, the influence of sex hormones on the immune response was found to play a role in the pathogenesis [26,27]. Whether the same might be true for infections with antibiotic-resistant bacteria needs to be investigated in future studies. 

In part, the gender differences in this study might be due to the pathogens themselves, since gender differences do not only apply to MRSA, CRA, and CRE. Several big studies showed higher incidences of *S. aureus* carriage and blood stream infections in males [9]. For the heterogenous group of Enterobacterales infections, gender preference depends on the site of infection and species involved [28]. One study from China found *Acinetobacter baumannii* infections to be more common among male hospital patients irrespective of specimen origin [29]. However, the proportion of carbapenem-resistance in *Acinetobacter* and *K. pneumoniae* isolates is still higher in males [8,30]. Whether the same or different mechanisms of gender-specific colonisation and infection apply to sensitive as to resistant bacteria should be further explored. 

Pig stables have been described to be a possible link for antibiotic-resistant bacteria between animals and humans [31]. We assessed whether men had an excessive risk for infection, due to exposure in this environment, as farmers or veterinarians in farm animal husbandry, by analysing the districts with the highest density of pigs and the most rural districts without observing a clear influence on the gender distribution. We also did not observe differing men-to-women IRR between the most and least deprived districts in Germany [32]. This suggests that regional and environmental factors, although they do affect AMR in many other ways, are probably not at the bottom of gender differences.

These findings are subject to several limitations. Our study design of a retrospective cohort study based on the German surveillance database does not account for people moving in and out of Germany, thereby altering the study population. However, in regard to gender distribution, the German population remained very stable over the ten years this study covers. Furthermore, although mandatory, surveillance data are not complete with respect to infections and colonisations with AMR pathogens in Germany. Additionally, notifications might vary between federal states due to differences in testing and diagnostic measures. Nevertheless, we do consider underreporting and other reporting-related biases as nondifferential between genders.

## 5. Conclusions

We provide estimates for the effect of gender on infections and colonisations with AMR pathogens that will be helpful in prediction models. Guidelines on screening procedures, as well as infection prevention and control measures, should take both age and gender of patients into account. For empiric antibiotic therapy, clinicians need to be aware that elderly men are an at-risk group for both infections and colonisations with AMR pathogens. Additional studies are necessary to clarify which underlying factors are drivers of the increased risk in men.

## Figures and Tables

**Figure 1 microorganisms-09-00894-f001:**
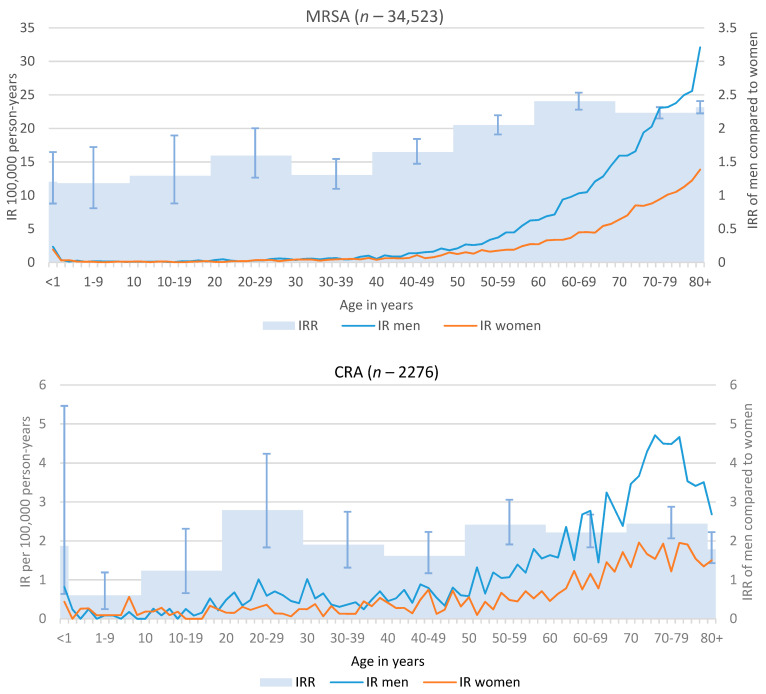
Incidence rates (IR) per 100,000 population by age and IR ratios (IRR) by age group for (**a**) invasive infection with Methicillin-resistant *S. aureus* (MRSA), 2010–2019, and infection or colonisation with carbapenem-non-susceptible (**b**) *Acinetobacter* spp. (CRA) and (**c**) Enterobacterales (CRE), 2017–2019.

**Table 1 microorganisms-09-00894-t001:** Baseline characteristics of notified cases of invasive infections with Methicillin-resistant *S. aureus* (MRSA), 2010–2019, and infections or colonisations with *Acinetobacter* spp. (CRA) and Enterobacterales (CRE), 2017–2019, in the German national surveillance database.

	MRSA *	CRA **		CRE **	
	All Notifications	All Notifications	Only Infections ***	All Notifications	Only Infections ***
	*n* (%)	*n* (%)	*n* (%)	*n* (%)	*n* (%)
Gender					
Female	12,510 (36.2)	781 (34.3)	191 (33.6)	4553 (37.9)	1076 (40.8)
Male	22,013 (63.8)	1495 (65.7)	377 (66.4)	7447 (62.1)	1564 (59.2)
Age group in years					
<1	162 (0.5)	15 (0.7)	2 (0.4)	352 (2.9)	31 (1.1)
1–9	109 (0.3)	28 (1.2)	3 (0.5)	202 (1.7)	46 (1.7)
10–19	108 (0.3)	40 (1.8)	7 (1.2)	154 (1.3)	29 (1.1)
20–29	313 (0.9)	118 (5.2)	21 (3.7)	434 (3.6)	95 (3.6)
30–39	546 (1.6)	127 (5.6)	21 (3.7)	536 (4.5)	97 (3.7)
40–49	1321 (3.8)	158 (6.9)	46 (8.1)	650 (5.4)	140 (5.3)
50–59	3585 (10.4)	342 (15.0)	87 (15.3)	1580 (13.1)	355 (13.4)
60–69	6594 (19.1)	496 (21.8)	138 (24.3)	2633 (21.8)	588 (22.2)
70–79	11,940 (34.6)	640 (28.1)	168 (29.6)	3277 (27.2)	710 (26.7)
80+	9873 (28.6)	314 (13.8)	75 (13.2)	2237 (18.6)	564 (21.2)
Federal state of residence					
Baden-Württemberg	1859 (5.4)	206 (9.0)	54 (9.5)	1316 (10.9)	297 (11.2)
Bavaria	2761 (8.0)	284 (12.5)	67 (11.8)	1575 (13.1)	350 (13.2)
Berlin	2407 (7.0)	268 (11.8)	73 (12.9)	1006 (8.4)	304 (11.5)
Brandenburg	1256 (3.6)	46 (2.0)	15 (2.6)	257 (2.1)	58 (2.2)
Bremen	254 (0.7)	14 (0.6)	4 (0.7)	87 (0.7)	31 (1.2)
Hamburg	431 (1.3)	116 (5.1)	31 (5.5)	378 (3.1)	82 (3.1)
Hessen	1874 (5.4)	300 (13.2)	40 (7.0)	1622 (13.5)	209 (7.9)
Lower Saxony	4427 (12.8)	133 (5.8)	37 (6.5)	612 (5.1)	107 (4.0)
Mecklenburg-Vorpommern	1122 (3.3)	18 (0.8)	7 (1.2)	109 (0.9)	31 (1.2)
North Rhine-Westphalia	10,654 (30.9)	587 (25.8)	157 (27.6)	2813 (23.3)	791 (29.8)
Rhineland-Palatinate	1076 (3.1)	79 (3.5)	13 (2.3)	577 (4.8)	81 (3.1)
Saarland	291 (0.8)	7 (0.3)	1 (0.2)	124 (1.0)	5 (0.2)
Saxony	2321 (6.7)	94 (4.1)	30 (5.3)	582 (4.8)	115 (4.3)
Saxony-Anhalt	1577 (4.6)	31 (1.4)	14 (2.5)	387 (3.2)	78 (2.9)
Schleswig-Holstein	1276 (3.7)	48 (2.1)	10 (1.8)	266 (2.2)	42 (1.6)
Thuringia	935 (2.7)	47 (2.1)	15 (2.6)	341 (2.8)	74 (2.8)
Year of notification					
2010	3754 (10.9)				
2011	4226 (12.2)				
2012	4487 (13.0)				
2013	4373 (12.7)				
2014	3850 (11.1)				
2015	3612 (10.5)				
2016	3183 (9.2)				
2017	2832 (8.2)	787 (34.6)	193 (34.0)	3444 (28.6)	695 (26.2)
2018	2433 (7.0)	780 (34.2)	196 (34.5)	3938 (32.7)	910 (34.3)
2019	1801 (5.2)	710 (31.2)	179 (31.5)	4673 (38.8)	1050 (39.6)
Hospitalisation status					
Inpatient	31,315 (90.6)	1905 (83.6)	493 (86.8)	10,329 (85.7)	2281 (85.9)
Outpatient	2026 (5.9)	227 (10.0)	50 (8.8)	979 (8.1)	290 (10.9)
Life status					
Deceased	3781 (10.9)	132 (5.8)	58 (10.2)	545 (4.5)	188 (7.1)
Not deceased	30,047 (87.0)	2119 (93.0)	506 (89.1)	11,362 (94.3)	2449 (92.2)

* MRSA study period: 2010–2019; ** CRA, CRE study period: 2017–2019; *** only infections represent a subgroup of all notifications for CRA and CRE. For MRSA, all notifications represent bloodstream infections.

**Table 2 microorganisms-09-00894-t002:** Multivariable analysis using Poisson regression of the ratio of infection/colonisation with AMR pathogens of men compared to women, stratified by infection vs. colonisation, hospitalisation status (inpatient vs. outpatient), and deceased vs. not deceased, Germany, 2010–2019.

	MRSA		CRA		CRE	
	Cr. IRR (95%CI)	Adj. IRR * (95%CI)	Cr. IRR (95%CI)	Adj. IRR * (95%CI)	Cr. IRR (95%CI)	Adj. IRR * (95%CI)
Infection			2.0 (1.7–2.4)	2.2 (1.9–2.7)	1.5 (1.4–1.6)	1.7 (1.6–1.8)
Colonisation			2.1 (1.8–2.3)	2.3 (2.0–2.6)	1.8 (1.7–1.9)	2.0 (1.9–2.1)
Inpatient	1.8 (1.7–1.8)	2.3 (2.2–2.3)	2.0 (1.8–2.2)	2.3 (2.0–2.5)	1.7 (1.6–1.8)	1.9 (1.9–2.0)
Outpatient	1.8 (1.7–2.0)	2.4 (2.1–2.6)	1.4 (1.1–1.8)	1.6 (1.2–2.0)	1.3 (1.1–1.4)	1.4 (1.3–1.6)
Deceased	1.6 (1.5–1.7)	2.2 (2.1–2.4)	1.8 (1.3–2.6)	2.1 (1.5–3.0)	1.8 (1.5–2.1)	2.2 (1.8–2.6)
Not deceased	1.8 (1.8–1.8)	2.4 (2.2–2.3)	1.9 (1.7–2.1)	2.1 (2.0–2.3)	1.6 (1.6–1.7)	1.9 (1.8–1.9)
All notifications	1.8 (1.7–1.8)	2.3 (2.2–2.3)	1.9 (1.8–2.1)	2.2 (2.0–2.4)	1.6 (1.6–1.7)	1.9 (1.8–2.0)

CI: Confidence interval, CRA: Carbapenem-non-susceptible *Acinetobacter* spp., CRE: Carbapenem-non-susceptible Enterobacterales, IRR: incidence rate ratio, MRSA: Methicillin-resistant *S. aureus.* * adjusted for age group, federal state of residence, and year of notification.

**Table 3 microorganisms-09-00894-t003:** Age-stratified hospitalisation rates (HR) of full-time inpatients per 100,000 population and HR ratios (HRR) of men compared to women in Germany, 2018.

Age Group	HR Men	HR Women	HRR (Men/Women)	Adjusted HR ^1^ Women	Adjusted HRR ^1^ (Men/Women)
<1	127,842	122,316	1.05	122,316	1.05
1–9	9376	7430	1.26	7430	1.26
10–19	8653	11,000	0.79	10,281	0.84
20–29	9558	18,238	0.52	9553	1.00
30–39	10,757	20,842	0.52	10,012	1.07
40–49	14,496	14,846	0.98	13,965	1.04
50–59	21,415	18,167	1.18	18,164	1.18
60–69	33,608	25,517	1.32	25,517	1.32
70–79	53,561	42,981	1.25	42,981	1.25
80+	71,558	62,643	1.14	62,643	1.14
**Overall**	23,121	24,523	0.94	22,038	1.05

^1^ Hospitalisation rates for women excluding hospitalisations due to pregnancy, birth, and postpartum.

## Data Availability

Aggregated data from a limited version of the German notification system database can be retrieved via SurvStat@RKI 2.0 https://survstat.rki.de/ (accessed on 15 March 2021). Detailed data are confidential and protected by German law and are available from the corresponding author upon reasonable request.

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
