# Peer review of "Bugs That Can Resist Antibiotics but Not Men: Gender-Specific Differences in Notified Infections and Colonisations in Germany, 2010–2019"

_microorganisms, 2021, doi:10.3390/microorganisms9050894_

Round 1
Reviewer 1 Report
Thank you for the opportunity to review the manuscript.
I recommend to improve the Introduction and Discussion section and formulate some Conclusions.
I believe more correlation could have been made in the Discussion section with the existing literature.
The resistance problem in highly sensitive and should be well documented.
The manuscript must be revised.
Reviewer 2 Report
This manuscript contributes a very interesting observation, which I must admit I was completely unaware of until I read the ms. However, I think the paper would improve a lot with some modifications
- Title is original but maybe too long. Perhaps “Bugs that can resist antibiotics but not men: Gender-specific differences in population-based incidence rates for notified in- fections and colonisations with antimicrobial-resistant bacteria in Germany, 2010-2019” may be replaced by something like “Bugs can resist antibiotics but not men: Gender-specific differences in infections and colonisations in Germany, 2010-2019”
- In table 1 legend the terms “All notifications” and “Only infections” should be clearly defined.
- I believe that data concerning infections caused by susceptible microorganisms and their gender distribution should be incorporated into the ms. It is necessary to show that the pohenomenon is exclusive to the resistant bacteria and not of a general nature.
Reviewer 3 Report
In this manuscript, Brandl et al showed that men in Germany had higher risk for infection with AMR pathogens than women. The manuscript is well structured and well written. I have only few comments as below,
Minor comments:
- line 56: Staphylococcus aureus should be italic.
- lines 132-133: the error should be corrected.
- Fig. 1 should be cited in the text.
- line 138: "ag group" should be "age group"
Round 2
Reviewer 1 Report
The manuscript is significantly improved.
Author Response
Response 1: We thank the reviewer for acknowledging the revision and improvement of the manuscript.
Reviewer 2 Report
I think the authors have introduced modifications adequately. Just two minor suggestions
- P6. legionella should be Legionella
2. Staphylococcus aureus should be spelled out the first tiume, then S. aureus
